# Separable roles for RNAi in regulation of transposable elements and viability in the fission yeast *Schizosaccharomyces japonicus*

**Elliott Chapman**[ID], **Francesca Taglini**[¤], **Elizabeth H. Bayne**[ID]*

Institute of Cell Biology, School of Biological Sciences, University of Edinburgh, Edinburgh, United Kingdom

¤ Current address: MRC Human Genetics Unit and CRUK Edinburgh Centre, Institute of Genetics and Cancer, University of Edinburgh, Edinburgh, United Kingdom

* elizabeth.bayne@ed.ac.uk

## Abstract

RNA interference (RNAi) is a conserved mechanism of small RNA-mediated genome regulation commonly involved in suppression of transposable elements (TEs) through both post-transcriptional silencing, and transcriptional repression via heterochromatin assembly. The fission yeast *Schizosaccharomyces pombe* has been extensively utilised as a model for studying RNAi pathways. However, this species is somewhat atypical in that TEs are not major targets of RNAi, and instead small RNAs correspond primarily to non-coding pericentromeric repeat sequences, reflecting a specialised role for the pathway in promoting heterochromatin assembly in these regions. In contrast, in the related fission yeast *Schizosaccharomyces japonicus*, sequenced small RNAs correspond primarily to TEs. This suggests there may be fundamental differences in the operation of RNAi pathways in these two related species. To investigate these differences, we probed RNAi function in *S. japonicus*. Unexpectedly, and in contrast to *S. pombe*, we found that RNAi is essential in this species. Moreover, viability of RNAi mutants can be rescued by mutations implicated in enhancing RNAi-independent heterochromatin propagation. These rescued strains retain heterochromatic marks on TE sequences, but exhibit derepression of TEs at the post-transcriptional level. Our findings indicate that *S. japonicus* retains the ancestral role of RNAi in facilitating suppression of TEs via both post-transcriptional silencing and heterochromatin assembly, with specifically the heterochromatin pathway being essential for viability, likely due to a function in genome maintenance. The specialised role of RNAi in heterochromatin assembly in *S. pombe* appears to be a derived state that emerged after the divergence of *S. japonicus*.

## Author summary

The chromosomes of many species are populated by repetitive transposable elements that are able to "jump" throughout the genome. The consequences of these mobilisations can be catastrophic, resulting in disruption of genes or chromosomal rearrangements, thus organisms usually employ defence mechanisms to keep these elements inactivated. The

**Data Availability Statement:** All RNA-Seq, sRNA-Seq and ChIP-Seq datasets have been deposited in the Gene Expression Omnibus (GEO) under accession number GSE185665. Genomic DNA

sequencing datasets have been deposited in the Sequence Read Archive (SRA) under accession number PRJNA770288.

**Funding:** This work was supported by a Wellcome Trust Investigator Award (202771/Z/16/Z; http://wellcome.org) and UK Medical Research Council Career Development Award (G1000505; http://mrc.ukri.org/) to EHB, and by the UK Biotechnology and Biological Sciences Research council (PhD studentships 1311387 to EC and 1101397 to FT; https://bbsrc.ukri.org). The funders had no role in study design, data collection and analysis, decision to publish, or preparation of the manuscript.

**Competing interests:** The authors have declared that no competing interests exist.

most widespread of these systems is RNA interference, which utilises small RNA molecules to direct either packaging of transposable element DNA into repressive heterochromatin, or degradation of RNA transcripts. Many fundamental discoveries about RNAi function have been made in the model fission yeast *Schizosaccharomyces pombe*; however, this species is unusual as it does not generally employ RNAi to control its transposable elements. We found that in a lesser studied relative, *Schizosaccharomyces japonicus*, small RNAs are required to silence transposable elements, and that this silencing occurs via both formation of heterochromatin and degradation of transcripts. This dual function RNAi pathway targeting transposable elements that appear to cluster at centromeres is very similar to systems seen in complex multicellular organisms, thus our findings reveal *S. japonicus* to be an exciting emergent model in which to study RNAi and centromere function.

## Introduction

RNA interference is believed to have evolved as an ancient defence mechanism against invasive genetic elements such as viruses and TEs [1]. While canonical RNAi pathways operate via small RNA-guided post-transcriptional silencing of target RNAs, in some cases small RNAs can also direct chromatin modification to enact transcriptional repression [2]. Unusually, in the fission yeast *S. pombe*, such chromatin-level regulation appears to be the dominant function of RNAi, with the majority of small RNAs corresponding to specialised pericentromeric repeat sequences that are assembled into heterochromatin in an RNAi-dependent manner. Non-coding pericentromeric transcripts are processed by the Dicer nuclease Dcr1 to generate short interfering (si)RNAs that guide the Argonaute effector protein Ago1 to complementary nascent RNAs [3–6]. This promotes recruitment of the histone methyltransferase Clr4 [7,8], resulting in methylation of lysine 9 on histone H3 (H3K9me) and heterochromatin assembly at cognate chromatin [2,9,10]. Loss of pericentromeric heterochromatin impairs centromere function and hence strains lacking core RNAi components such as Dcr1 or Ago1, while viable, exhibit defects in chromosome segregation [11–13].

While the *S. pombe* RNAi system has become a paradigm for small RNA-directed chromatin modification, it appears atypical of RNAi pathways in general as it retains limited capacity for canonical post-transcriptional silencing [14–16], and plays a largely redundant role in TE control [17–20]. *S. pombe* harbours only two closely related TE families, Tf1 and Tf2, belonging to the group of long terminal repeat (LTR) retrotransposons that includes the Ty3 element of *Saccharomyces cerevisiae* and the Gypsy element of *Drosophila melanogaster* [21]. Tf element-derived sequences make up around 1% of the sequenced *S. pombe* genome, and most of these are LTR remnants, with only 13 full-length copies of Tf2 and none of Tf1 [22]. These elements are dispersed throughout euchromatic regions of the genome, and are regulated by a specialised pathway involving CENP-B-like proteins that bind specific DNA sequences in LTRs and repress transcription [23–25]. TE transcripts are also subject to elimination by the exosome, and only in certain conditions, such as when exosome-mediated degradation is impaired, are TE transcripts strongly targeted by RNAi generating siRNAs that direct heterochromatin assembly at TE sequences [20].

*S. japonicus* is another member of the fission yeast clade, estimated to have diverged from *S. pombe* around 220 million years ago [23]. Recent comparative analyses have revealed notable differences in genome organisation between the two species; in particular, *S. japonicus* appears to lack the specialised pericentromeric repeat sequences found in *S. pombe*, but has a larger

complement of retrotransposons (comprising at least 2% of the genome) that appear to cluster at putative centromeric and telomeric loci [23,26]. The assemblies of these regions are incomplete, however eleven distinct families have been identified thus far, named Tj1 –Tj11. These all belong to the Ty3/Gypsy family, but show much greater diversity than elements in *S. pombe*, falling into two major lineages, one (including Tj1, Tj4 and Tj6) related to Tf1/2 and the other (including Tj2, Tj3 Tj5, Tj7, Tj8 Tj9 Tj10 and Tj11) related to Ty3. Moreover, it appears there must also be key differences in mechanisms of retrotransposon regulation, since the CENP-B-like pathway that is active in *S. pombe* is absent in *S. japonicus* [23,27]. Interestingly however, retrotransposons in *S. japonicus* give rise to abundant siRNAs [23], and are enriched for heterochromatin-associated H3K9 methylation [23,26], suggesting that, in contrast to *S. pombe*, TE regulation may represent a major function of RNAi in *S. japonicus*.

In order to investigate functional divergence in RNAi pathways within the fission yeast clade, we set out to dissect the role of RNAi in *S. japonicus*. Unexpectedly, we were generally unable to recover viable deletion mutants for RNAi factors, indicating that unlike in *S. pombe*, RNAi is essential for viability in *S. japonicus*. However, two viable $dcr1^+$ deletion strains were recovered, and analysis of these revealed that viability was linked to compensatory mutations promoting maintenance of H3K9 methylation independently of RNAi. While high levels of H3K9 methylation were maintained over retrotransposon loci in these strains, retrotransposons were nevertheless deregulated at the post-transcriptional level. Our findings reveal that *S. japonicus* retains the ancestral role of RNAi in regulating retrotransposons, and that the pathway serves dual functions: heterochromatin assembly over retrotransposons that is essential for viability, and post-transcriptional silencing of these elements that is dispensable.

## Results

To investigate RNAi function in *S. japonicus* we began by attempting to construct strains bearing deletions of genes encoding core RNAi factors including the sole Dicer and Argonaute proteins, Dcr1 and Ago1. We also attempted to delete genes encoding homologues of other factors critical for RNAi in *S. pombe* including Rdp1, Stc1, Chp1, Arb1 and Arb2. Unexpectedly, with the exception of rare $dcr1^+$ mutant isolates (see below), we were unable to recover deletion mutants for any of these factors (Figs 1A and S1). We were also unable to recover strains deleted for the H3K9 methyltransferase Clr4. This was in contrast to deletion of genes known to not be required for the maintenance of heterochromatin ($pku70^+$, $pku80^+$ and $tri1^+$) [28], which could be achieved with high efficiency (S1 Fig). Integration of epitope tags at RNAi gene loci could also be achieved with similar high efficiency (S1 Fig), indicating that failure to recover RNAi deletion mutants does not reflect an intrinsic problem in targeting these loci. Rather, these observations suggest that, unlike in *S. pombe*, a functional RNAi pathway is essential for viability in *S. japonicus*. To verify this conclusion, we tested whether an RNAi gene could be deleted following genomic insertion of a second copy of the gene at an ectopic locus. Whereas we were unable to delete the $ago1^+$ gene in wild-type cells, deletions of endogenous $ago1^+$ were readily recovered in the presence of an ectopic copy of the gene that maintained Ago1 function (Fig 1A). Similar results were obtained for $dcr1^+$ and $clr4^+$. These findings strongly suggest that, in contrast to *S. pombe*, a functional RNAi pathway is usually required for cell viability in *S. japonicus*.

After extensive screening, we were able to recover two viable isolates bearing deletions of the $dcr1^+$ gene (hereafter denoted as $dcr1\Delta^*$ and $dcr1\Delta^†$ to indicate that these strains constitute rare survivors that may have acquired secondary mutations). The phenotypes of the two strains were indistinguishable, and for clarity we focus on results for the $dcr1\Delta^*$ strain (data for the $dcr1\Delta^†$ strain are shown in S10 Fig). Northern blot analysis indicated that retrotransposon-

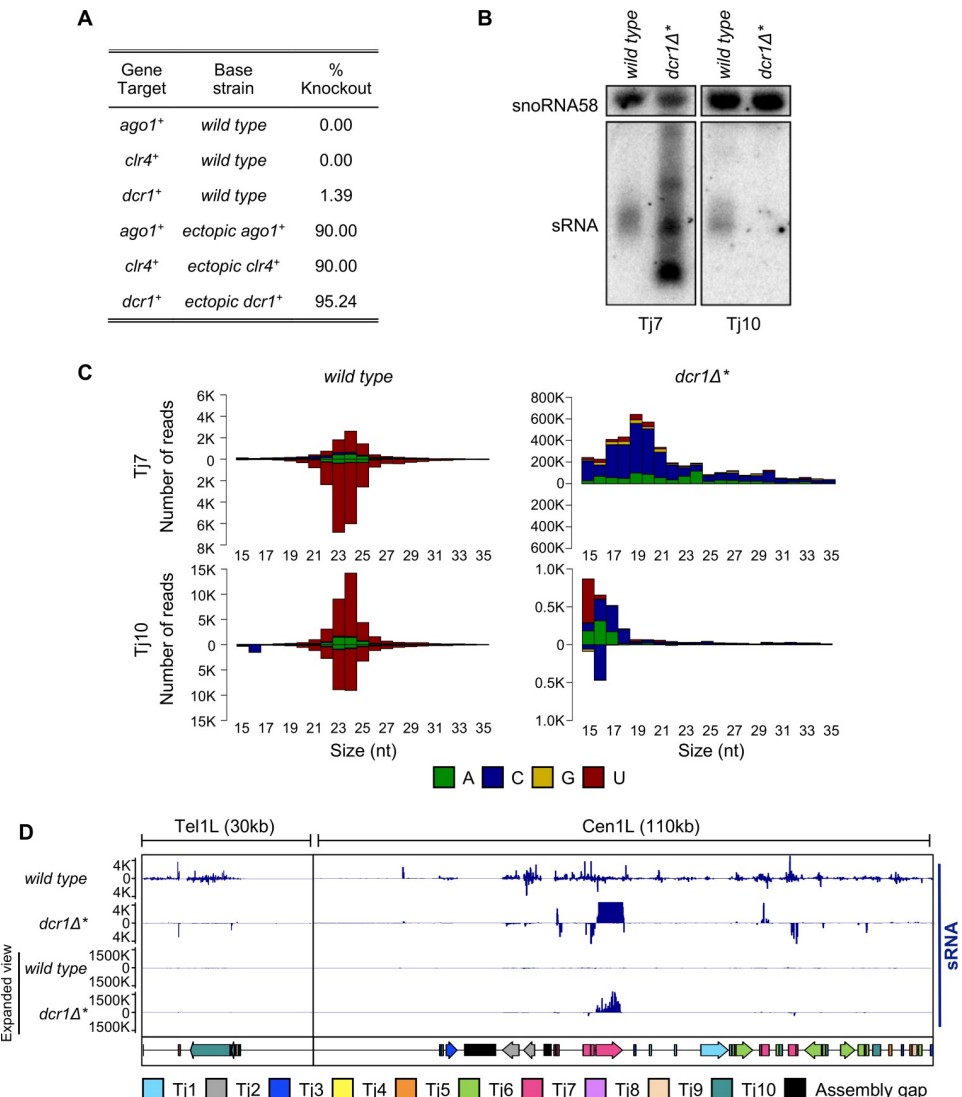

**Fig 1. A functional RNAi pathway is required for viability in *S. japonicus*.** **(A)** Knockout recovery rates of core RNAi and heterochromatin factors in wild-type and ectopic expression backgrounds. **(B)** Northern blot analysis of small RNA species isolated from wild-type and *dcr1Δ\** strains, probed with ³²P end-labelled oligonucleotides, antisense to the indicated retrotransposon or snoRNA58 loading control. **(C)** Size profile, strand bias and 5' nucleotide preference of small RNA species that map to indicated retrotransposons, isolated from wild-type and *dcr1Δ\** strains. RNAs derived from the sense strand are plotted above the axis, whilst RNAs derived from the antisense strand are plotted below. **(D)** Genome view of mapped small RNA species at two retrotransposon-rich loci, in wild-type and *dcr1Δ\** backgrounds. Note differing scale in the expanded view tracks. The positions of individual transposable elements are indicated by coloured arrows below the data track.

derived small RNA populations were lost or altered in the deletion strains, consistent with the loss of function of Dcr1 (Figs 1B and S2). This was further confirmed by small RNA-seq analysis, which revealed that populations of predominantly 23–24 nucleotide long siRNAs mapping to all ten families of Tj retrotransposon in wild-type cells were lost in the *dcr1Δ\** mutant (Figs 1C and S3). For some families of Tj element such as Tj7, discrete populations of small RNAs were detectable in the *dcr1Δ\** strain; however, these did not have the characteristic size distribution or 5'U bias associated with bona fide siRNAs (Figs 1C and S3) [29,30]. Moreover, whereas siRNAs are typically derived from both strands, these Dcr1-independent small RNAs

corresponded almost exclusively to the sense strand of the retrotransposons, suggesting they are generated via an alternative RNA processing pathway (Fig 1C and 1D).

We reasoned that the increase in abundance of these alternative, sense-derived small RNA species in the *dcr1Δ\** mutant cells may reflect elevated retrotransposon mRNA levels. Consistent with this, RNA-seq analysis revealed increased accumulation of transcripts corresponding to centromere and telomere-clustered retrotransposons in *dcr1Δ\** compared to wild-type cells (Fig 2A). Indeed, TEs were found to constitute the majority of upregulated loci in *dcr1Δ\** cells (Fig 2B). Increased retrotransposon expression was also confirmed by RT-qPCR, with Tj7 showing the highest levels of transcript accumulation (Figs 2C and S4A). These observations indicate that, in contrast to *S. pombe* [18,19], Dcr1 plays a major role in regulating TE expression in *S. japonicus*.

To assess whether the increased retrotransposon expression is also associated with increased transposition, we assessed element copy number by qPCR in *dcr1Δ\** versus wild-type, parental cells. Nine out of ten retrotransposon families showed no increase in copy number in *dcr1Δ\** cells, suggesting that despite increased mRNA levels, these elements are not competent for transposition. On the other hand, Tj7, which showed the highest levels of transcript accumulation in the absence of functional Dcr1, increased in copy number by approximately 10-fold (Figs 2D and S4B). Coupled with the observation that Tj7 was the source of the vast majority of Dcr1-independent small RNAs in *dcr1Δ\** cells, (Fig 1B and 1C), this suggests that these alternative small RNA species may be associated with the retrotransposition process. Indeed, it is known that in the reverse transcription step of the mobilisation cycle, copying of single-stranded RNA into DNA is followed by RNase H-mediated cleavage of the RNA in the resulting DNA-RNA heteroduplex [31]. There is also evidence to suggest that RNase H cleavage sites are selected based on nucleotide preference within a defined sequence window [32–34]. Thus the observed Tj7-derived Dcr1-independent small RNA species are most likely RNase H cleavage products.

The RNA-seq analysis also revealed a number of genes up- and down-regulated in *dcr1Δ\** cells (S1 Table). However, Gene Ontology analysis did not reveal any particular pathway enrichment amongst these genes, nor any other factors implicated in TE regulation. Interestingly, we also noted a number of previously unannotated genomic loci associated with increased transcript and decreased siRNA levels in the *dcr1Δ\** cells, a signature otherwise characteristic of retrotransposon sequences. Indeed, analysis of these regions using the NCBI Conserved Domain Search tool [35] revealed that these loci largely comprised sequences encoding retrotransposon-specific protein domains (an example locus is shown in Fig 2E). Multiple-sequence alignment of these regions allowed us to identify 10 new retrotransposon families, that we have named Tj12 to Tj21 (S2 Table). Phylogenetic analyses indicated that each of the new TEs falls into one of the two existing retrotransposon lineages in *S. japonicus* (S5 Fig). The majority of TEs in the *S. japonicus* genome are densely clustered in putative centromeric and telomeric regions, and the newly identified retrotransposons tend to reside in windows of previously unannotated sequence within these regions, consistent with *S. japonicus* centromeres being composed almost exclusively of retrotransposon-derived sequences.

In *S. pombe*, the RNAi pathway mediates silencing primarily at the chromatin level, being required for proper maintenance of H3K9 methylation on pericentromeric repeats. To investigate whether silencing of retrotransposons is mediated via a similar mechanism in *S. japonicus*, we analysed H3K9 methylation by ChIP-seq. Unexpectedly, we found high levels of H3K9 methylation at retrotransposons in both wild-type and *dcr1Δ\** cells (Figs 3A and S6). This was confirmed by ChIP-qPCR analyses, which showed no significant change in H3K9me2 levels at any retrotransposons in *dcr1Δ\** cells, suggesting that Dcr1 is not required to maintain H3K9me2 at TEs in these cells (Figs 3B and S7A). That H3K9 methylation levels are unaffected

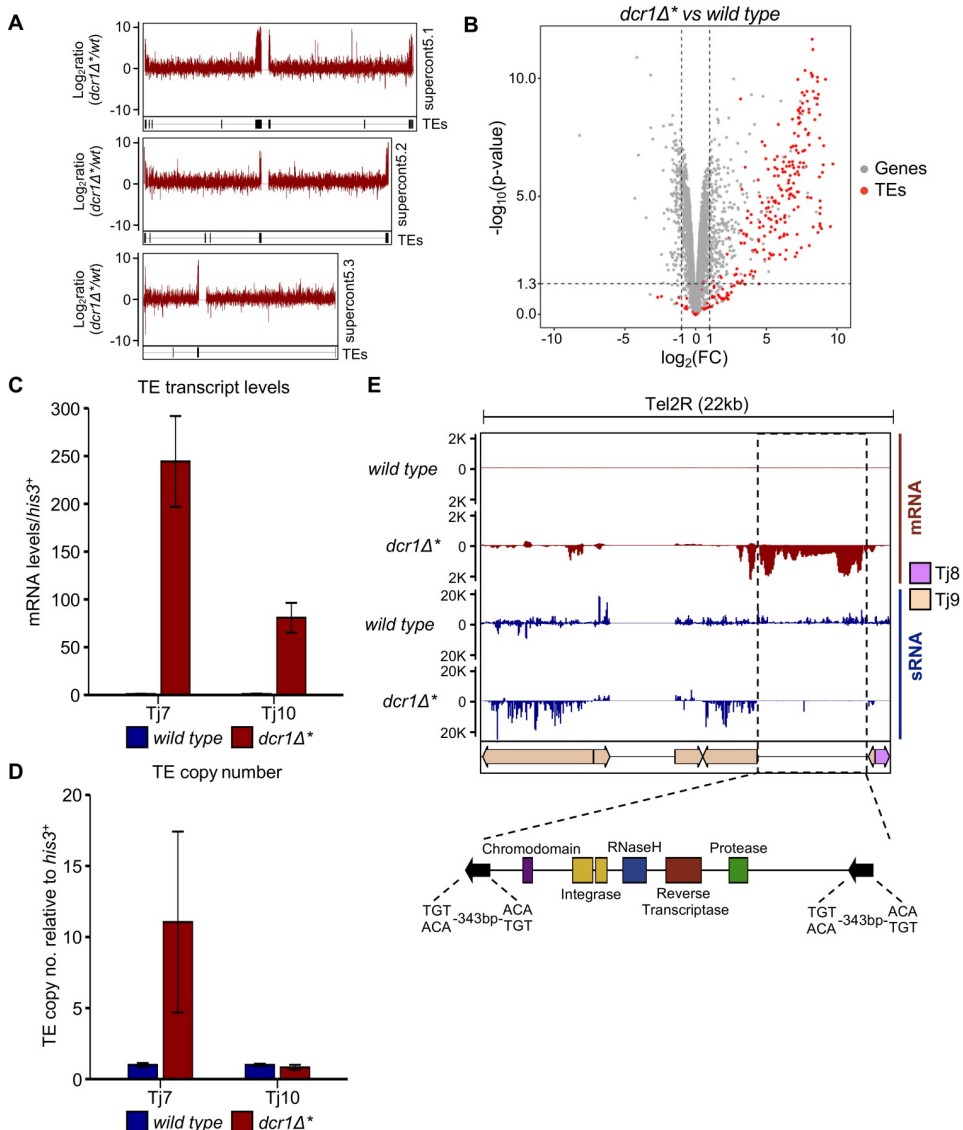

**Fig 2. Rare *dcr1Δ* survivors exhibit activation of transposable elements.** **(A)** Ratio of mapped RNA-Seq reads in *dcr1Δ\** vs wild-type cells for the three main chromosomal contigs of the SJ5 annotation. TE locations are indicated by black bars underneath each data track. **(B)** Volcano plot of differentially expressed genes from RNA-Seq in *dcr1Δ\** vs wild-type cells. Differentially expressed retrotransposons are indicated in red, with all other genes in grey. Dotted lines indicate a–log10(p-value) of ~1.3 (equivalent to p = 0.05) and -1 > Log2FC > 1 (indicating a 2-fold change in expression in mutant vs wild-type). **(C)** RT-qPCR analysis of retrotransposon transcript levels, relative to *his3+*, normalised to wild-type. Data plotted are the mean ± SD from three replicates. **(D)** qPCR analysis of retrotransposon copy number, relative to *his3+*, normalised to wild-type. Data plotted are the mean ± SD from three replicates. **(E)** Example of a novel TE locus discovered from analysis of siRNA and mRNA-seq data. Regions that displayed loss of small RNAs and gain of mRNA signal in the *dcr1Δ\** mutant vs wild-type were searched for known retrotransposon protein motifs using the NCBI domain search tool. These sequences were then searched against the rest of the SJ5 genome sequence using BLAST to identify the location of other TE copies. The positions of individual transposable elements are indicated by coloured arrows below the data track.

in *dcr1Δ\** cells suggests that the observed increase in TE transcript accumulation is not due to loss of heterochromatin-mediated transcriptional silencing. As a complementary approach to verify this, we also assessed RNA polymerase II (PolII) association with retrotransposon loci by RNA PolII ChIP-qPCR, the expectation being that if increased transcript accumulation is

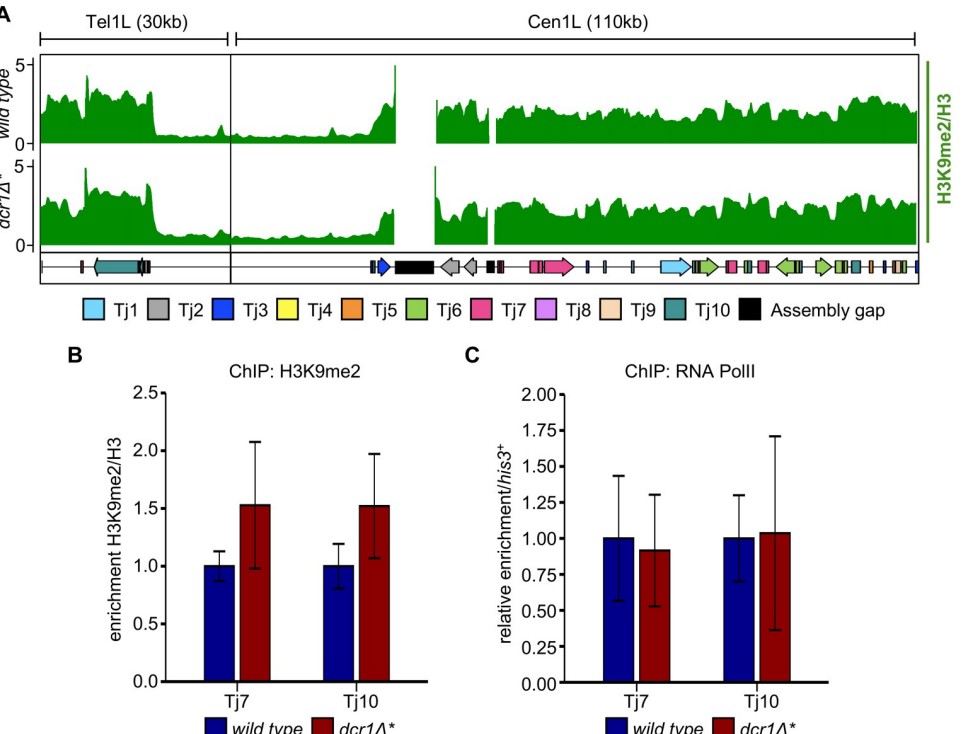

**Fig 3. Levels of H3K9 methylation and transcription at retrotransposon loci are unaffected in rare *dcr1Δ* survivors.** **(A)** ChIP-seq profile of H3K9me2 across two retrotransposon-rich loci in wild-type and *dcr1Δ\** cells. Enrichments are given in reads per kilobase million (RPKM) and represent the ratio of input normalised anti-H3K9me2 precipitated chromatin over input normalised anti-H3 precipitated chromatin. The positions of individual transposable elements are indicated by coloured arrows below the data track. **(B)** ChIP-qPCR analysis of H3K9me2/H3 levels at Tj7 and Tj10 retroelements, relative to *his3⁺*, normalised to wild-type. Data plotted are the mean ± SD from three replicates. **(C)** ChIP-qPCR analysis of RNA PolII levels at Tj7 and Tj10 retroelements, relative to *his3⁺*, normalised to wild-type. Data plotted are the mean ± SD from three replicates.

due to transcriptional upregulation, this will be associated with increased RNA PolII occupancy. However, consistent with the H3K9me2 ChIP results, we observed no change in RNA PolII occupancy at retrotransposons in *dcr1Δ\** cells (Figs 3C and S7B), indicating that increased transcript accumulation is not a result of increased transcription. Together these observations indicate that the elevated retrotransposons transcript levels observed in *dcr1Δ\** cells must be largely the result of loss of RNAi-mediated post-transcriptional silencing.

Since RNAi appears to be essential for viability in *S. japonicus*, and *dcr1Δ* strains were recovered only at very low frequency, we reasoned that some compensatory mutation(s) may be enabling viability in these rare *dcr1Δ* isolates. To search for such mutations, we undertook genome-resequencing of the two *dcr1* deletion strains. Sequence analysis revealed only a small number of mutations affecting protein-coding sequences, and from these a single strong candidate mutation was identified in each of the mutant strains. One strain (*dcr1Δ\**) carried a missense mutation (R77W) in the essential gene *mpe1⁺*, which encodes the ortholog of budding yeast Mpe1, an E3 ubiquitin ligase that is a component of the cleavage and polyadenylation factor (CPF) complex [36,37] (S8A Fig). This is interesting since recent studies have implicated other components of the CPF complex in RNAi-independent heterochromatin assembly in *S. pombe* [38–40]. Hence one possibility is that the R77W mutation creates a gain of function allele of *mpe1⁺* that enhances H3K9 methylation in the absence of Dcr1. The other strain (*dcr1Δ†*) carried a frameshift mutation in the gene encoding Leo1, a component of the RNA

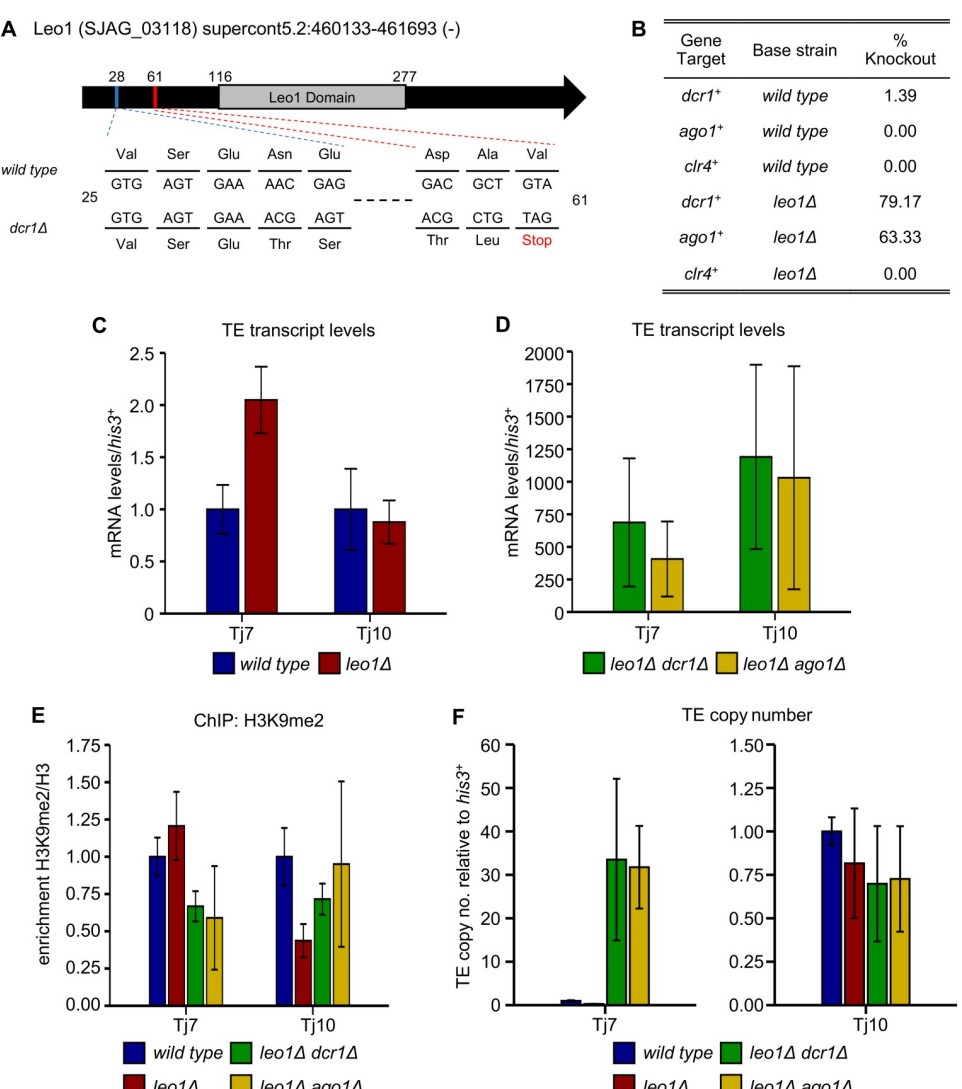

**Fig 4. Mutation of the Paf1 complex subunit Leo1 facilitates survival of RNAi mutants. (A)** Schematic showing the location of the sequenced mutation within Leo1. The single base pair deletion causes a frameshift, which leads to the introduction of a premature stop codon after 61 amino acids. **(B)** Knockout recovery rates of core RNAi and heterochromatin factors in wild-type and *leo1Δ* backgrounds. **(C)** RT-qPCR analysis of retrotransposon transcript levels in wild-type and *leo1Δ* strains, relative to *his3+*, normalised to wild-type. Data plotted are the mean ± SD from three replicates. **(D)** RT-qPCR analysis of retrotransposon transcript levels in *leo1Δ dcr1Δ* and *leo1Δ ago1Δ* strains, relative to *his3+*, normalised to wild-type. Data plotted are the mean ± SD from three replicates. **(E)** ChIP-qPCR analysis of H3K9me2/H3 levels at Tj7 and Tj10 retroelements, relative to *his3+*, normalised to wild-type. Data plotted are the mean ± SD from three replicates. **(F)** qPCR analysis of retrotransposon copy number, relative to *his3+*, normalised to wild-type. Data plotted are the mean ± SD from three replicates.

polymerase-associated factor (PAF) complex (S8B Fig). The frameshift is predicted to introduce a premature stop codon at position 61, resulting in a severely truncated protein (Fig 4A). This mutation was particularly striking since we and others have recently shown that Leo1 is a negative regulator of heterochromatin that antagonises H3K9 methylation in *S. pombe* [41,42]. Thus loss of Leo1 function might be expected to facilitate maintenance of H3K9 methylation in the absence of RNAi.

To confirm whether loss of Leo1 function is sufficient to rescue the lethality associated with *dcr1+* deletion in *S. japonicus*, we generated a new *leo1Δ* strain, and then attempted to delete

*dcr1*+ in this background. Strikingly, isolates deleted for *dcr1*+ could be recovered with high efficiency in the *leo1Δ* background (79% positive, Fig 4B). This efficiency was almost as high as when a second copy of *dcr1*+ was present in the background (95% positive, Fig 1A). Thus absence of Leo1 can largely compensate for loss of Dcr1. To further investigate the mechanism of compensation, we next tested whether *leo1*+ deletion can also suppress lethality associated with deletion of two other factors: the RNAi component Ago1, or the H3K9 methyltransferase Clr4. Interestingly, *ago1Δ* mutants could also be recovered with high efficiency in the *leo1Δ* background, suggesting that deletion of Leo1 can compensate for the loss of not just Dcr1, but the RNAi pathway more generally (Figs 4B and S1). In contrast, we were unable to recover *clr4*+ deletion mutants in the *leo1Δ* background. This indicates that compensation is dependent on H3K9 methylation. These observations are consistent with a model whereby lethality in RNAi mutants is associated with loss of H3K9 methylation, and deletion of *leo1*+ restores viability by maintaining H3K9 methylation independently of RNAi. ChIP-qPCR analyses confirmed that H3K9me2 levels at retrotransposons remain high in *leo1Δ dcr1Δ* and *leo1Δ ago1Δ* double mutants, similar to wild-type cells (Figs 4E and S9A). In contrast, retrotransposons are still de-regulated in these mutants, as indicated by elevated transposon transcript levels and copy numbers relative to parental cells (Figs 4C, 4D, 4F, S9B and S9C). The double mutant strains also display growth defects relative to wild-type cells that are comparable to those in the original *dcr1Δ* survivors (S10A Fig), which may reflect the deleterious effects of increased retrotransposon mobilisation. These findings indicate that RNAi targets retroelements at two distinct levels in *S. japonicus*: chromatin regulation through H3K9 methylation, and post-transcriptional regulation independent of H3K9 methylation. The maintenance of H3K9 methylation appears to be essential for viability, whilst post-transcriptional regulation of retroelements does not.

## Discussion

Suppression of TEs is an ancient and conserved role of small RNAs in eukaryotes. However, in the widely studied fission yeast model, *S. pombe*, the RNAi pathway has acquired a specialised role in heterochromatin assembly at specific repeat sequences and plays only a minor role in retrotransposon control in normal conditions [17–20]. Our observations now indicate that this specialisation occurred after the divergence from *S. japonicus*, the earliest branching member of the fission yeast clade, since an important function for RNAi in repressing active TEs is retained in *S. japonicus*. This repression occurs at both the transcriptional level, via deposition of repressive histone marks over retrotransposon sequences, and at the post transcriptional level, via Argonaute-mediated degradation of retrotransposon transcripts. This co-ordinated utilisation of transcriptional and post-transcriptional silencing in TE control is reminiscent of RNAi pathways in multicellular eukaryotes, and also has parallels with piRNA mediated repression of TEs in animals [43,44]. In these cases, whilst transcriptional silencing occurs in the nucleus, post-transcriptional transcript cleavage takes place in the cytoplasm, mediated by distinct Argonaute/PIWI family proteins [44]. As *S. japonicus* encodes only a single Argonaute, it will be interesting to investigate whether this protein forms distinct complexes for transcriptional and post-transcriptional repression, and whether these similarly function in discrete subcellular compartments.

Interestingly, we note that different families of TEs display different responses to the loss of RNAi-mediated regulation in *S. japonicus*. For example, high levels of RNA accumulation were observed for both Tj7 and Tj10; however, only for Tj7 did this appear to be associated with increased transposition. The reason for this difference is currently unclear: we can find full-length copies of most of the identified TEs including Tj10, so there is no conspicuous

reason why Tj10 would be non-functional, and conversely, there are no additional features in the sequence of Tj7 that might account for why this family alone is competent for transposition. A current limitation is that the precise genomic location of the majority of TEs in *S. japonicus* remains unknown due to the poor assembly of centromeric and telomeric regions; it is possible that resolving the position and chromatin context of these elements may shed light on the apparent differences in their function/regulation. Related to this, we do not know where in the genome new copies of Tj7 are integrated in the d*cr1Δ** cells; however, we note that Tj7 contains a chromodomain, a common feature of Ty3 type elements that has been shown to direct their integration into heterochromatic loci [45,46]. We therefore suspect that additional copies of Tj7 will have integrated into existing heterochromatic TE arrays. Also related, the relatively high levels of transcript accumulation for elements such as Tj7 in the d*cr1Δ** cells are perhaps somewhat surprising given that these elements are nevertheless associated with H3K9 methylation that is expected to repress transcription. However, it is possible that not all copies of these elements are equally enriched for H3K9me2, and some copies are efficiently repressed, while others are expressed (the high level of sequence identity precludes distinguishing between individual elements in the ChIP and expression data). Another, non-mutually exclusive, possibility is that this form of heterochromatin is permissive for transcription (as has been reported for H3K9me2-marked heterochromatin in *S. pombe* [47]).

An unexpected finding is that H3K9 methylation is essential for viability in *S. japonicus*. In mammals, depletion of RNAi factors Dicer or Ago2 leads to embryonic lethality [48,49], whilst disruption of Suv39h1/h2 (the mammalian homologues of Clr4) severely reduces viability during embryogenesis [50]. However, *S. pombe* strains lacking RNAi or heterochromatin factors are viable. It is possible that uncontrolled TE activity may contribute to the lethality associated with loss of H3K9 methylation in *S. japonicus*; however, since TE dysregulation is also observed in viable *leo1Δ dcr1Δ* cells, we suspect that this may relate more to a critical role for H3K9 methylation in supporting centromere function. In *S. pombe*, H3K9 methylation is required for binding of the HP1 protein Swi6 at pericentromeres, which in turn recruits cohesin to mediate tight sister chromatid cohesion and support proper interactions between kinetochores and the mitotic spindle [51–53]. Loss of Swi6 or H3K9 methylation causes loss of centromeric cohesion resulting in elevated rates of chromosome loss associated with increased incidence of lagging chromosomes on late anaphase spindles [51,54,55]. However, presence of lagging chromosomes is also associated with slowed spindle elongation, suggesting that *S. pombe* cells possess a mechanism that can sense a lagging chromosome and create extra time for it to reach the pole [56]. The nature of this mechanism is not understood, but if it is absent in *S. japonicus*, this could account for the much more severe defects in mutants lacking pericentromeric H3K9 methylation. In addition, while centromeric cohesin is lost in *S. pombe* heterochromatin mutants, arm cohesin is unaffected, and it appears that residual arm cohesion can partially compensate for the loss of centromeric cohesion and maintain the fidelity of chromosome segregation at a level compatible with viability [51,57]. Therefore another possibility is that in *S. japonicus*, arm cohesion alone may be less effective in supporting accurate chromosome segregation, such that loss of centromeric heterochromatin has more catastrophic effects. Future comparative analyses of the factors underlying the differential sensitivity to loss of heterochromatin in these yeasts may shed further light on the evolution of mechanisms supporting centromere function.

Whilst the centromeric regions in *S. pombe* comprise specialised repeat sequences, in *S. japonicus* centromeres are defined by arrays of retrotransposon sequence. This is reminiscent of arrangements seen in humans, where centromeres contain both retroviral elements and long-interspersed nuclear elements (LINEs) [58], and plants, where specific centromeric retrotransposons (CRs) have been defined that are targeted by RNAi [59]. Likewise, Drosophila

centromeres are TE-rich [60,61], and processing of retrotransposon transcripts into siRNAs is required to maintain pericentromeric heterochromatin [62]. Thus in the utilisation of RNAi as a primary mechanism to regulate TEs, the centromeric clustering of these mobile elements, and their use as a platform for assembly of critical heterochromatin, *S. japonicus* has much in common with plants and animals, establishing it as an attractive model in which to study both RNAi and centromere function.

## Materials and methods

### Strains, media, and culture conditions

*S. japonicus* strains used in this study are listed in S3 Table. Cells were grown in rich liquid YES media [63], or plated onto rich solid YES or minimal solid PMG media at 32˚C unless otherwise stated. Growth assays were performed by plating serial 10-fold dilutions of selected strains onto solid YES media and incubating at 32˚C for 4 days.

### Strain construction

As genomic integration of exogenous DNA into *S. japonicus* requires long flanking homologies [64], fragments for homologous recombination were either amplified from purpose-built vectors or were constructed using overlap fusion PCR. For the deletion of $ago1^+$, $clr4^+$ and $dcr1^+$, vectors were built by restriction endonuclease cloning, using pFA6a-natMX6 as a backbone (all plasmids used in this study are listed in S4 Table). For the $ago1\Delta$ and $clr4\Delta$ constructs, flanking homologies were generated by PCR amplification of a 1kb region upstream of the target gene start codon and a 1kb region downstream of the target gene stop codon, these were then subsequently restriction cloned into pFA6a-natMX6, either side of the natMX6 cassette. For the $dcr1\Delta$ constructs, the upstream flanking homology was generated in the same way as for $ago1\Delta$ and $clr4\Delta$, whilst two versions of the downstream flanking homology were constructed, both of which delete a portion of the $dcr1^+$ open reading frame without interfering with any genes coded on the opposite strand. The first deletes the initial 108bp of the coding sequence and was named $dcr1\Delta^{dis}$, whilst the second deletes a larger 2520bp portion and was termed $dcr1\Delta^{del}$. These flanking homologies were also restriction cloned into pFA6a-natMX6, either side of the natMX6 cassette. The resulting four vectors: pFA6a-$dcr1\Delta^{dis}$-natMX6, pFA6a-$dcr1\Delta^{del}$-natMX6, pFA6a-$ago1\Delta$-natMX6 and pFA6a-$clr4\Delta$-natMX6 were used as templates for PCR. Deletion constructs for $arb1^+$, $arb2^+$, $chp1^+$, $rdp1^+$, $rik1^+$, $stc1^+$, $tri1^+$, $pku70^+$ and $pku80^+$ were constructed using overlap fusion PCR of three fragments generated by PCR amplification of a 1kb region upstream of the target gene start codon, the HindIII $ura4^+$ fragment from *S. pombe*, and a 1kb region downstream of the target gene stop codon. For deletion of $leo1^+$ a vector was built using the NEBuilder HiFi kit, whereby a 1kb upstream homology arm, the HindIII $ura4^+$ fragment from *S. pombe* and a 1kb downstream homology arm were cloned into a pFA6a backbone. The resulting vector pFA6a-$leo1\Delta$-ura4 was used as a template for PCR. In order to introduce ectopic gene copies at the $nmt1^+$ locus, an *S. japonicus* version of the pREP1 vector was first constructed, whereby the *S. pombe* $nmt1^+$ promoter and terminator were swapped for the corresponding *S. japonicus* versions [65]. The open reading frames of $ago1^+$, $clr4^+$ and $dcr1^+$ were subsequently cloned between the $nmt1^+$ promoter and terminator, and the resultant plasmids (pREP1SJ-$ago1^+$, pREP1SJ-$clr4^+$, pREP1SJ-$dcr1^+$) were used as templates for PCR. For C-terminal tagging of $chp1^+$, $rik1^+$ and $stc1^+$, 1kb fragments corresponding to the region immediately upstream of the target gene stop codon, and the region downstream of the target gene 3'UTR, were PCR amplified and cloned into either pSO729 (eGFP) [66] or pFA6a-FLAG-natMX6 (5xFLAG) [67]. The resultant plasmids were linearised with NdeI for pFA6a-FLAG-NatMX6 or XhoI for pSO729 for use as linear transformation fragments. N-

terminal tagging of *ago1*[+] was achieved using overlap fusion PCR of three fragments comprising a 1kb region upstream of the *ago1*[+] start codon, the kanMX6-P3nmt1-NFLAG module from pFA6a-kanMX6-P3nmt1-NFLAG [67], and the first 1kb of the *ago1*[+] coding sequence. Transformation of *S. japonicus* was performed by electroporation [68], and cells were selected on the appropriate YES + antibiotic or PMG amino acid dropout plates.

## RNA isolation

Total RNA was extracted from 2ml of cells at $OD_{595}$ = 0.8–1.0 using the Masterpure Yeast RNA Purification Kit (Lucigen). Small RNAs were extracted by resuspending 50ml of pelleted cells at $OD_{595}$ = 0.8–1.0 in 50 mM Tris–HCl pH 7.5, 10 mM EDTA pH 8, 100 mM NaCl, 1% SDS, and adding equal volumes of phenol:chloroform 5:1 and acid washed beads. Cells were lysed using a bead beater (Biospec products) for 2 x 2 min at 4˚C. The soluble fraction was extracted with phenol/chloroform and long RNAs precipitated with 10% polyethylene glycol 8000 and 0.5 M NaCl on ice for 30 min. The supernatant was recovered and small RNAs precipitated with ethanol overnight at −20˚C.

## Northern blot

Northern analysis of small RNAs was performed as described previously [69]. Briefly, RNA samples were run on a 12% polyacrylamide gel, electrophoretically transferred onto Hybond-NX (Amersham) and crosslinked by incubation at 55˚C for 2 hours with a 0.16 M carbodiimide, 1-ethyl-3-(3-dimethylaminopropyl) carbodiimide (EDC) solution. Membranes were probed with 5′ end radiolabelled oligonucleotides listed in S5 Table.

## RT-qPCR

For RT-qPCR analysis, 1µg of total RNA was treated with TURBO DNase (Ambion) for 1 hour at 37˚C, then reverse transcribed using random hexamers (Roche) and Superscript III reverse transcriptase (Invitrogen) according to the manufacturer's instructions. cDNA was quantified by qPCR using LightCycler 480 SYBR Green (Roche) and primers listed in S5 Table.

## RNA-seq

Libraries were prepared using Illumina TruSeq stranded mRNA library preparation kit according to manufacturer's instructions, then pooled and sequenced paired-end on an Illumina HiSeq4000. Bioinformatic analysis was carried out using the Galaxy server [70]. Raw reads were filtered for quality and adapter using Trim Galore! (version 0.4.3.1) [71]. Trimmed reads were then aligned to the *S. japonicus* reference genome SJ5 (GCA_000149845.2) using STAR (version 2.7.2b) [72]. Read counts were obtained using featureCounts (version 1.6.4) [73] with settings -M–fraction [74], and differential gene expression then performed using edgeR (version 3.21.1) [75]. For visualisation, bedgraph coverage files were generated using bamCoverage (version 3.3.2.0.0) [76]. Gene Ontology (GO) and KEGG pathway analysis was performed using goseq (version 1.44.0) [77].

## Small RNA-seq

Libraries were prepared using lllumina TruSeq small RNA library preparation kit according to manufacturer's instructions, then pooled and sequenced single-end on an Illumina HiSeq2500. Bioinformatic analysis was carried out using the Galaxy server [70], Raw reads were filtered for quality and adapter using Trim Galore! (version 0.4.3.1) [71]. Trimmed reads

were then aligned to the *S. japonicus* reference genome SJ5 (GCA_000149845.2) using bowtie2 (version 2.3.4.3) [78]. For visualisation, bedgraph coverage files were generated using bamCoverage (version 3.3.2.0.0) [76]. Plots detailing the length, strand bias and first nucleotide preference of small RNAs mapping to transposable elements were created using a previously published custom R script [79].

## Annotation of new retrotransposon families

Read coverage tracks for the mRNA-Seq and sRNA-Seq experiments were loaded into IGV alongside the latest SJ5.gff file (Schizosaccharomyces_japonicus. GCA_000149845.2.51.gff3), updated to include the co-ordinates of the retrotransposons Tj1-Tj10, as previously specified [23]. Regions that did not have an annotation but exhibited lost or changed small RNA signals and increased mRNA coverage in the *dcr1Δ\** mutant were manually identified. The NCBI Conserved Domain Search tool [35] was then used to search these regions for any sequences potentially coding for retrotransposon-associated protein domains (gag, protease, reverse transcriptase, RNase, integrase, chromodomain). Where such domains were found, up- and downstream sequences were searched for direct repeats that could correspond to LTRs, which define retrotransposon borders and would indicate the presence of a full length TE. Any regions larger than 1kb that contained at least one retrotransposon-related protein motif were then searched against rest of the *S. japonicus* reference genome SJ5 (GCA_000149845.2) to identify any additional copies. From this analysis, three previously unannotated full length retrotransposons, 79 partial retrotransposons, and 32 solo-LTRs were identified. Alignment of the newly identified reverse transcriptase, RNase H and integrase domain sequences with the corresponding domains from Tj1-10 using Clustal Omega revealed 10 novel retrotransposon families (elements with greater than 95% sequence identity within these domains were considered to be from the same family). We also identified the Tj11 element recently described by the Allshire lab [26]. The co-ordinates of the retrotransposons Tj1-21 are detailed in S2 Table.

## Phylogenetic analysis of retrotransposons

In order to classify the newly discovered retrotransposons, the sequences of the reverse transcriptase, RNase and integrase domains were aligned to the corresponding domains from Tj1-10, as well as *S. pombe* Tf1 and Tf2 and *S. cerevisiae* Ty3 using ClustalW [80]. IQtree2 [81] was then used to build a single tree based on the three loci, with 1000 ultrafast bootstraps. IQtree2 selected TVM+F+R3 as the best-fitting base substitution model using the Bayesian Information Criterion (BIC). The resulting tree was opened and edited using FigTree [82] to display the bootstrapping values as node labels.

## ChIP-qPCR

50ml of cells per IP were grown in YES to an $OD_{595}$ of 0.8–1.0, and were fixed in 1% formaldehyde for 15 min at room temperature. Cells were lysed in 350μl of 50mM Hepes-KOH pH7.5, 140mM NaCl, 1mM EDTA, 1% (v/v) Triton X-100, 0.1%(w/v) sodium deoxycholate, 1x PMSF, 1x yeast protease inhibitors (Roche) using a bead beater (Biospec products) for 2 x 2 min at 4˚C. Sonication was performed using a Bioruptor Twin (Diagenode) for a total of 20 min (30 sec on/30 sec off on 'high' power). Samples were pre-cleared using 25μl Protein G Agarose beads, before lysates were incubated overnight at 4˚C, with 1μl per IP of anti-H3K9me2 (mAb 5.1.1 [83]), 2μl per IP of anti-H3 (abcam ab1791), or 5μl per IP of anti-PolII (Sigma-Aldrich 8WG16) and 25μl Protein G Agarose beads. Beads were washed with Lysis Buffer, Lysis Buffer with 500 mM NaCl, wash buffer (10 mM Tris-HCl pH 8, 0.25 M LiCl, 0.5% NP-40, 0.5% (w/v) sodium deoxycholate,1 mM EDTA) and TE. Immunoprecipitated DNA

was recovered by first boiling with Chelex-100 resin (BioRad) for 12 min, before incubating with Proteinase K (Roche) for 30 min at 55°C with shaking at 1000rpm. Quantification of enrichment was performed by qPCR using LightCycler 480 SYBR Green (Roche) and primers listed in S5 Table. Relative enrichments were calculated as the ratio of product of interest to control product (*his3$^+$*) in IP over input. In all cases, histograms represent three biological replicates and error bars represent one S.D.

### ChIP-seq

150ml of cells per IP were grown in YES to an OD$_{595}$ of 0.8–1.0 and fixed in 1% formaldehyde for 15 min at room temperature. Cells were lysed in 650μl of 50mM Hepes-KOH pH7.5, 140mM NaCl, 1mM EDTA, 1% (v/v) Triton X-100, 0.1%(w/v) sodium deoxycholate, 1x PMSF, 1x yeast protease inhibitors (Roche) using a bead beater (Biospec products) for 4 x 2 min at 4°C. Sonication was performed in lysis buffer containing 0.2% SDS, using a Bioruptor Twin (Diagenode) for a total of 28 min (30 sec on/30 sec off on 'high' power). Samples were pre-cleared using 50μl Protein G Agarose beads, before lysates were incubated overnight at 4°C, with 3μl per IP of anti-H3K9me2 (mAb 5.1.1 [83]) or 6μl per IP of anti-H3 (abcam ab1791) and 75μl Protein G Agarose beads. Beads were washed twice each with Lysis Buffer, Lysis Buffer with 500 mM NaCl, wash buffer (10 mM Tris-HCl pH 8, 0.25 M LiCl, 0.5% NP-40, 0.5% (w/v) sodium deoxycholate,1 mM EDTA) and TE. Immunoprecipitated DNA was recovered by incubating beads with 1x Elution buffer (10mM Tris-HCl pH8.0, 300mM NaCl, 5mM EDTA, 1% SDS) or input sample with 1.5x Elution buffer (15mM Tris-HCl pH8.0, 450mM NaCl, 7.5mM EDTA, 1.5% SDS) at 65°C for at least 6 hours with shaking at 1000rpm. Samples were then treated with RNase A at 37°C for 1 hour, before incubation with Proteinase K at 55°C for 2 hours. Immunoprecipitated DNA was recovered using PCR purification columns (Qiagen). ChIP-Seq libraries were prepared using NEBNext Library Prep Master Mix Set (NEB) according to manufacturer's instructions, then pooled and sequenced paired-end on an Illumina HiSeq 4000. Bioinformatic analysis was carried out using the Galaxy server [70], Raw reads were filtered for quality and adapters using Trim Galore! (version 0.4.3.1) [71]. Trimmed reads were then aligned to the *S. japonicus* reference genome SJ5 (GCA_000149845.2) using bowtie2 (version 2.3.4.3) [78]. bamCompare (version 3.3.2.0.0) [76] was used to compute the ratio of the number of reads between IP and Input samples.

### Genomic DNA isolation

Genomic DNA was extracted from 10ml of cells grown to early stationary phase. Cells were lysed in SPZ buffer (1.2M Sorbitol, 50mM sodium citrate, 50mM Na$_2$HPO$_4$, 40mM EDTA, 400μg/ml Zymolyase 100T (Amsbio), adjusted to pH 5.6) at 37°C, and genomic DNA extracted by adding 10% SDS/5M KOAc. DNA was precipitated using isopropanol and contaminating RNA removed using RiboShredder RNase Blend (Epicentre). RNase was removed using an equal volume of phenol:chloroform:isoamyl alcohol (25:24:1), and DNA was reprecipitated with ethanol for 10 min at -80°C.

### gDNA-Seq

For sequencing of genomic DNA, libraries were prepared using Illumina Nextera XT DNA Library Prep Kit according to manufacturer's instructions, then pooled and sequenced paired-end on an Illumina NextSeq550. Bioinformatic analysis was carried out using the Galaxy server [70]. Raw reads were filtered for quality and adapters using Trim Galore! (version 0.4.3.1). SNPs were called using the Snippy package (version 4.5.0) [84] with *S. japonicus* SJ5 (GCA_000149845.2) as the reference genome.

## Supporting information

**S1 Fig. Core RNAi and heterochromatin factors cannot be deleted without the presence of an ectopic gene copy or a suppressor mutation, but they can be epitope tagged.** Knockout and tagging rates of targeted genes in the wild-type background, in the presence of an ectopic gene copy, or in a *leo1Δ* background.
(TIF)

**S2 Fig. Deletion of Dcr1 abolishes siRNA production, however specific transposable elements give rise to alternative small RNA species.** Northern blot of small RNA species isolated from wild-type and *dcr1Δ\** strains, probed with $^{32}$P end-labelled oligonucleotides, antisense to the indicated retrotransposon or snoRNA58, a loading control.
(TIF)

**S3 Fig. TE derived small RNA species present in the absence of Dcr1 lack the characteristics of siRNAs.** Size profile, strand bias and 5' nucleotide preference of small RNA species that map to indicated retrotransposons, isolated from wild-type and *dcr1Δ\** strains. RNAs derived from the sense strand are plotted above the axis, whilst RNAs derived from the antisense strand are plotted below.
(TIF)

**S4 Fig. Absence of Dcr1 is associated with accumulation of retrotransposon transcript for a majority of TE families, but mobilisation of only Tj7. (A)** RT-qPCR analysis of retrotransposon transcript levels, relative to *his3$^+$*, normalised to wild-type. Data plotted are the mean ± SD from three replicates. Data for Tj7 and Tj10 are the same as in Fig 2C. **(B)** qPCR analysis of retrotransposon copy number, relative to *his3$^+$*, normalised to wild-type. Data plotted are the mean ± SD from three replicates. Data for Tj7 and Tj10 are the same as in Fig 2D.
(TIF)

**S5 Fig. Phylogenetic analysis of newly discovered retrotransposons indicates that these are novel elements that fall into the two existing lineages *of S. japonicus* retroelements.** For each newly discovered retrotransposon, the sequences of the reverse transcriptase, RNase and integrase domains were aligned to the corresponding domains from Tj1-10, as well as *S. pombe* Tf1 and Tf2 and *S. cerevisiae* Ty3 using ClustalW (Tj19 was excluded as we could find only partial elements in which these domains were not present). IQtree2 was then used to build a single tree based on the three domains, with 1000 ultrafast bootstraps. IQtree2 selected TVM+F+R3 as the best-fitting base substitution model using the Bayesian Information Criterion (BIC). Node labels indicate bootstrapping values. Scale bar is in units of base substitutions per site. Retrotransposons discovered in this study (Tj12 –Tj21) are highlighted in blue, previously discovered retrotransposons (Tj1 –Tj11) are highlighted in red.
(TIF)

**S6 Fig. Genome wide H3K9 methylation levels are unaffected in the absence of Dcr1.** ChIP-seq profile of H3K9me2 in *dcr1Δ\** vs wild-type cells for the three main chromosomal contigs of the SJ5 annotation. Enrichments are given in reads per kilobase million (RPKM) and represent the ratio of input normalised anti-H3K9me2 precipitated chromatin over input normalised anti-H3 precipitated chromatin. TE locations are indicated by black bars underneath each data track.
(TIF)

**S7 Fig. H3K9me2 levels and RNA PolII occupancy at retrotransposons are unchanged in the absence of Dcr1. (A)** ChIP-qPCR analysis of H3K9me2/H3 levels across retrotransposons,

normalised to *his3*$^+$. Data plotted are the mean ± SD from three replicates. **(B)** ChIP-qPCR analysis of RNA PolII levels across retrotransposons. *his3*$^+$ is included to illustrate RNA PolII occupancy at a transcribed gene. Data plotted are the mean ± SD from three replicates.
(TIF)

**S8 Fig. Rare *dcr1Δ* survivors carry potential compensatory mutations that facilitate their survival. (A)** Table of SNPs within coding regions sequenced from the *dcr1Δ*$^*$ strain. **(B)** Table of SNPs within coding regions sequenced from the *dcr1Δ*$^†$ strain.
(TIF)

**S9 Fig. *leo1Δ dcr1Δ and leo1Δ ago1Δ* strains show similar activation of retrotransposons independent of H3K9 methylation. (A)** ChIP-qPCR analysis of H3K9me2/H3 levels across retrotransposons, normalised to *his3*$^+$. Data plotted are the mean ± SD from three replicates. **(B)** RT-qPCR analysis of retrotransposon transcript levels, relative to *his3*$^+$, normalised to wild-type. Data plotted are the mean ± SD from three replicates. Data for Tj7 and Tj10 are the same as in Fig 4D & 4E. **(C)** qPCR analysis of retrotransposon copy number, relative to *his3*$^+$, normalised to wild-type. Data plotted are the mean ± SD from three replicates. Data for Tj7 and Tj10 are the same as in Fig 4F.
(TIF)

**S10 Fig. The *dcr1Δ*$^†$ strain exhibits similar growth and molecular phenotypes to the *dcr1Δ*$^*$ strain. (A)** Growth assay showing the growth phenotypes of the rare *dcr1Δ* survivor strains and the *leo1Δ dcr1Δ* and *leo1Δ ago1Δ* strains, in comparison to wild-type and *leo1Δ* strains. **(B)** RT-qPCR analysis of retrotransposon transcript levels, relative to *his3*$^+$, normalised to wild-type. Data plotted are the mean ± SD from three replicates. **(C)** qPCR analysis of retrotransposon copy number, relative to *his3*$^+$, normalised to wild-type. Data plotted are the mean ± SD from three replicates. **(D)** ChIP-qPCR analysis of H3K9me2/H3 levels across retrotransposons, normalised to wild-type. Data plotted are the mean ± SD from three replicates.
(TIF)

**S1 Table. List of up- and down- regulated genes in *dcr1Δ*$^*$ vs wild-type cells.**
(XLSX)

**S2 Table. Genomic coordinates of the retrotransposon elements Tj1-21.**
(XLSX)

**S3 Table. *S. japonicus* strains used in this study.**
(DOCX)

**S4 Table. Plasmids used in this study.**
(DOCX)

**S5 Table. Oligonucleotides used in this study.**
(DOCX)

**S1 Data. Underlying numerical data for all graphs.**
(XLSX)

## Acknowledgments

Thanks are due to the Oliferenko lab for strains and advice on *S. japonicus* transformation, to Hironori Niki for strains, and to Takeshi Urano for the H3K9me2 antibody. Thanks to Darren Obbard for advice on construction of phylogenetic trees. We are grateful to Alison Pidoux and Jo Strachan for advice and comments on the manuscript and to members of the Bayne lab for

discussions. Total RNA and siRNA sequencing was carried out by Edinburgh Genomics, The University of Edinburgh. ChIP and genomic DNA sequencing was carried out by the Edinburgh Clinical Research Facility.

## Author Contributions

**Conceptualization:** Elliott Chapman, Francesca Taglini, Elizabeth H. Bayne.

**Data curation:** Elliott Chapman.

**Formal analysis:** Elliott Chapman.

**Funding acquisition:** Elizabeth H. Bayne.

**Investigation:** Elliott Chapman, Francesca Taglini.

**Methodology:** Elliott Chapman, Francesca Taglini, Elizabeth H. Bayne.

**Supervision:** Elizabeth H. Bayne.

**Visualization:** Elliott Chapman.

**Writing – original draft:** Elliott Chapman, Elizabeth H. Bayne.

**Writing – review & editing:** Elliott Chapman, Francesca Taglini, Elizabeth H. Bayne.

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
