## [Decision Letter · Decision Letter 0]

6 Dec 2021

Dear Dr Bayne,

Thank you very much for submitting your Research Article entitled 'Separable roles for RNAi in regulation of transposable elements and viability in the fission yeast Schizosaccharomyces japonicus' to PLOS Genetics.

The manuscript was fully evaluated at the editorial level and by three independent peer reviewers who are all experts in the field.  The reviewers appreciated the attention to an important topic but identified some concerns that we ask you address in a revised manuscript. All three indicated that this constitutes minor revision. There are several places where they ask for additional clarification or documentation, so we ask that you pay careful attention to these points.

We therefore ask you to modify the manuscript according to the review recommendations. Your revisions should address the specific points made by each reviewer.

[LINK]

Please let us know if you have any questions while making these revisions.  We very much appreciate your submitting this exciting and important study to PLOS Genetics, and look forward to receiving your revised manuscript.

Yours sincerely,

Joe

Joseph Heitman, MD, PhD

Associate Editor

PLOS Genetics

John Greally

Section Editor: Epigenetics

PLOS Genetics

Reviewer's Responses to Questions

**Comments to the Authors:**

Reviewer #1: The study reported the study of the RNA interference pathway in Schizosaccharomyces japonicus. This yeast belongs to the fission yeast clade of which S. pombe is the most studied member. The authors report that the RNAi pathway is essential in J. japonicus in contrast to the situation in S. pombe. Still, they were able to identify rare viable dcr1delta transformants. In this background the cells do no produce siRNA and the level of expression of retrotransposon is increased so is the transposition of a least one them. In contrast, the authors observed no modification of the H3K9me2 marks at these retrotransposon loci. The analysis of genome of two of these rare dcr1dleta mutant revealed that they are suppressors bearing a mutation at the MPE1 and LEO1 locus, respectively. Overall, this study reveals the dual role of RNAi in S. japonicus controlling both the chromatin structure and the TE expression through a post-trancriptional pathway. This is well done study and a well written manuscript. Yet, several points need to be clarified as explained below:

1) Fig1B: the Northern indeed revealed that absence of TJ10 small RNAs but might suggest that it is not the case for TJ7. It is the RNA-Seq that revealed that the signal observed in the TJ7 Northern are not siRNA. So the sentence saying “Northern blot analysis indicated that retro-transposon-derived siRNAss were lost…” is false and should be modified.

2) The authors reported the identification of dcr1delta suppressor mutant but not ago1delta equivalent. Although they deleted AGO1 in the leo1 background. It is possible to do in the dcr1delta* mutant bearing the MPE1 mutation?

3) The authors revealed an increased in the number of copies of TJ7 in the drc1* mutant. Where are these copies added in the genome? They did not observed any increase in the copy number of TJ10. Why these TJ10 retrotransposons do not transpose? Are they nonfunctional or truncated in some ways?

4) What is the level of expression of TE in the leo1 background?

5) According to the presented model, in the dcr1* mutant the only post-transcriptional repression pathway controlling TE expression is affected but still the expression of TJ7 increased by 200 fold in the presence of “normal” H3K9me2 marks. Could the authors comment on this apparent very poor efficiency of H3K9me2 marks to control gene expression in this yeast?

Reviewer #2: It is well established that RNAi is a conserved mechanism that can suppress transposition of transposable elements (TEs) via transcriptional and post-transcriptional mechanisms. Interestingly, loss of RNAi in the model fission yeast Schizosaccharomyces pombe does not result in major defects in TE silencing, as the exosome RNA degradation machinery rapidly degrades transcripts from those elements to prevent their reactivation. In this manuscript, Chapman et al. probe the contribution of RNAi to TE regulation in Schizosaccharomyces japonicus, an organism that is closely related to S. pombe. Surprisingly, the authors discover and comprehensively demonstrate that RNAi and heterochromatin are essential for viability in S. japonicus, in contrast to what has been found for S. pombe. Importantly, the mutation or loss of Leo1, a subunit of the Paf1 complex, allows RNAi mutants to survive. H3K9me2 levels at TEs in such strains are unaffected but reactivation can occur, as some TEs are upregulated and one of them (i.e. Tj7) can transpose. In summary, the authors propose that RNAi regulates TEs in S. japonicus by two mechanisms, a transcriptional one involving heterochromatin formation and a post-transcriptional one involving degradation. Overall, the manuscript is well written and convincing. I would recommend it for publication in PLoS Genetics provided the following modifications are included:

Introduction/Discussion:

As mentioned, the authors must acknowledge that RNAi indeed plays an important role in controlling retrotransposons in S. pombe, but such a role is masked by the exosome RNA degradation machinery (Yamanaka et al., Nature 2012 PMID: 23151475). This raises the intriguing possibility that the exosome may not be targeting retrotransposons in S. japonicus, therefore rendering the role of RNAi essential.

Figure 3A, ChIP-seq:

Please add the genome-wide data as a Supp. Figure, so the reader can judge the enrichment at heterochromatic regions in comparison to euchromatin.

Figs. 3B-C and S6, ChIP-qPCR:

While it is very clear that there is no difference in the enrichment levels between WT and various dcr1∆ mutants, the absence of negative controls makes it difficult to assess the quality of the ChIPs. Please show the raw unprocessed data as a Supp. Figure or Supp. Table.

Figure 2D:

I wonder what it is the biological consequence of having a higher number of Tj7 copies and increased transcription of various Tjs in RNAi mutants. Could the authors provide serial dilution assays or liquid growth curves to assess the fitness of those strains (i.e. rare survivors and leo1∆ RNAi∆ double mutants)?

Uniqueness of Tj7

Is there any unique motif/s in the Tj7 sequence that might explain why this TE, but not other elements, is able to transpose?

Figure S5:

The authors state “Phylogenetic analyses indicated that each of the new TEs falls into one of the two existing retrotransposon lineages in S. japonicus (Figure S5)”. However, Tj19 is not displayed in Figure S5. This must be clarified. Please also state in the figure legend that Tj12-21 elements were identified in this study, whereas Tj11 was already known.

The description about the existence of RNAi-independent heterochromatin assembly pathway shall include a reference to article by Reyes-Turcu et al (PMID: 21892171). Similarly, reference to Vo et al 2019 (PMID: 31269446) shall be included (along with ref 36) when describing the requirement of CPF complex in RNAi-independent heterochromatin assembly

Reviewer #3: Chapman and colleagues explore the role of RNAi in the fission yeast S. japonicus. This work is interesting on several grounds. As the authors highlight, RNAi has been investigated in the more intensely studied fission yeast S. pombe. The lineage leading to S. pombe has, however, acquired an RNAi-independent transposon defense pathway (not found in S. japonicus) and no longer relies heavily on RNAi for this role. S. japonicus, therefore, has a lot of potential as a model for studying RNAi. S. japonicus retains much of the tractability of S. pombe and the authors demonstrate that the S. japonicus RNAi pathway retains a role more like the ancestral state of the pathway shared with many other eukaryotes. I thought the paper was clear, accessible and the claims were mostly supported. I enjoyed reading this.

My one hesitation to feel the claims were fully reported was that the role for leo1 in S. japonicus has not been demonstrated. I agree with the authors that it is most likely the gene does the same thing in S. japonicus as it does in S. pombe. However, given the significant differences between the two species demonstrated in this work and the long time since they shared a common ancestor, it would be nice to have more support for the idea that Leo1 antagonizes H3K9 methylation in S. japonicus. The persistent inability to get clr4 KO even in a leo1 background helps convince me that the authors’ interpretations are correct, but figure 4B is not supported by a table with the number of transformants screened. Those data are provided for Figure 1A and I found that important.

I have some additional suggestions listed below. These are not major deficiencies but are mostly suggestions I think could help improve the paper.

Overall, I think more details about the japonicus transposons are warranted. I suggest adding more to introduce the transposon landscape in japonicus. How much more transposon dense is japonicus than pombe? What types of transposons are there (e.g. TF-like, Ty-like)? This was answered in the results, but I was wondering about this earlier in the paper. I was also confused by the naming system of japonicus transposons. I suggest the authors describe what determines if a transposon sequence represents a new family rather than a member of a previously defined family. Is there a formal cutoff of divergence that separates families? I was also wondering if there was any information about the mobility of the different transposon families, given they did not respond the same way to the mutations assayed in this paper.

Were all assays done for the two dcr1 deletion mutants? The text says the phenotypes were indistinguishable, but that claim was hard to evaluate. Could those data be provided as a supplement?

Some more contextual information to introduce the motivation and expectations in the PolIII CHiP qPCR could help general audience readers.

In the discussion, the term ‘higher eukaryotes’ was used. I suggest rewording to multicellular eukaryotes.

The centromeric cohesin/chromosome segregation model for the H3K9 methylation requirement seems reasonable. Is it also possible that too much transposon activity could contribute to the lethality associated with loss of H3K9 methylation?

Figure 3A. Please provide more info about the scale.

Figure S1: There are several genes not mentioned in the paper. Folks may miss them.

For some of the experiments, I think the same experiments shown in the main text and supplemental figure. I appreciate that as a reader as it makes comparisons easier. I suggest, however, mentioning in the legend of the supplement if this is the same data shown again.

For Figure S5, what are the ‘three loci’? There seem to be many more than three considered in the tree.

Figure S8: The letters are missing from the figure.

**Have all data underlying the figures and results presented in the manuscript been provided?**

Reviewer #1: Yes

Reviewer #2: Yes

Reviewer #3: **No: **It is specified in the review that the data for the one of the dcr1 deletions is not provided. There also needs to be a table specifying the number of colonies screened for the data underlying figure 4B

PLOS authors have the option to publish the peer review history of their article (what does this mean?). If published, this will include your full peer review and any attached files.

Reviewer #1: No

Reviewer #2: No

Reviewer #3: No

---

## [Decision Letter · Decision Letter 1]

14 Feb 2022

Dear Elizabeth,

We are pleased to inform you that your revised PLOS Genetics manuscript entitled "Separable roles for RNAi in regulation of transposable elements and viability in the fission yeast Schizosaccharomyces japonicus" has been editorially accepted for publication in PLOS Genetics. Congratulations!

Thank you again for supporting open-access publishing; we are looking forward to publishing your work in PLOS Genetics and appreciate your entrusting this exciting study to PLOS and its broad readership!

Yours sincerely,

Joe

Joseph Heitman, MD, PhD

Associate Editor

PLOS Genetics

John Greally

Section Editor: Epigenetics

PLOS Genetics

Comments from the reviewers (if applicable):

Reviewer's Responses to Questions

**Comments to the Authors:**

Reviewer #1: no further question

Reviewer #2: The manuscript by Chapman et al. has been further improved and clarified as the authors have addressed all my concerns and added suggested references. Indeed, I am more confident in the ChIP-qPCR and ChIP-seq experiments as proper controls and genome-wide data are now shown, respectively. In addition, the authors now show that various RNAi mutants (e.g. dcr1∆ survivors and leo1∆ RNAi∆ doubles) exhibit reduced fitness. This finding is important as it illustrates the biological consequences of having higher number of copies and increased transcription of Tjs. Taken together, I am happy to recommend this ms. for publication in PLoS Genetics.

Reviewer #3: The authors have sufficiently addressed all my concerns and comments. This is a solid paper and i enjoyed reading it again.

**Have all data underlying the figures and results presented in the manuscript been provided?**

Reviewer #1: Yes

Reviewer #2: Yes

Reviewer #3: Yes

PLOS authors have the option to publish the peer review history of their article (what does this mean?). If published, this will include your full peer review and any attached files.

Reviewer #1: No

Reviewer #2: No

Reviewer #3: No

**Data Deposition**

http://datadryad.org/submit?journalID=pgenetics&manu=PGENETICS-D-21-01431R1

**Press Queries**

---

## [Editor Report · Acceptance letter]

23 Feb 2022

PGENETICS-D-21-01431R1 

Separable roles for RNAi in regulation of transposable elements and viability in the fission yeast *Schizosaccharomyces japonicus*

Dear Dr Bayne, 

We are pleased to inform you that your manuscript entitled "Separable roles for RNAi in regulation of transposable elements and viability in the fission yeast *Schizosaccharomyces japonicus*" has been formally accepted for publication in PLOS Genetics! Your manuscript is now with our production department and you will be notified of the publication date in due course.

With kind regards,

Zsofia Freund

PLOS Genetics

On behalf of:
